# Glycomacropeptide in PKU—Does It Live Up to Its Potential?

**DOI:** 10.3390/nu14040807

**Published:** 2022-02-14

**Authors:** Anne Daly, Alex Pinto, Sharon Evans, Anita MacDonald

**Affiliations:** Birmingham Women’s and Children’s Hospital NHS Foundation Trust, Steelhouse Lane, Birmingham B4 6NH, UK; alex.pinto@nhs.net (A.P.); evanss21@me.com (S.E.); anita.macdonald@nhs.net (A.M.)

**Keywords:** glycomacropeptide, PKU, protein substitute, amino acids

## Abstract

The use of casein glycomacropeptide (CGMP) as a protein substitute in phenylketonuria (PKU) has grown in popularity. CGMP is derived from κ casein and is a sialic-rich glycophosphopeptide, formed by the action of chymosin during the production of cheese. It comprises 20–25% of total protein in whey products and has key biomodulatory properties. In PKU, the amino acid sequence of CGMP has been adapted by adding the amino acids histidine, leucine, methionine, tyrosine and tryptophan naturally low in CGMP. The use of CGMP compared to mono amino acids (L-AAs) as a protein substitute in the treatment of PKU promises several potential clinical benefits, although any advantage is supported only by evidence from non-PKU conditions or PKU animal models. This review examines if there is sufficient evidence to support the bioactive properties of CGMP leading to physiological benefits when compared to L-AAs in PKU, with a focus on blood phenylalanine control and stability, body composition, growth, bone density, breath odour and palatability.

## 1. Introduction

It is estimated there are 0.45 million people worldwide with the inherited metabolic disorder phenylketonuria (PKU) [1], which causes irreversible neurological damage if untreated. Although pharmaceutical therapies are being actively developed, a phenylalanine restricted diet remains the only effective treatment. In classical PKU, protein substitutes (low phenylalanine protein replacements) provide up to 80% of dietary protein requirements and are essential to ensure metabolic stability and growth. Protein substitutes are derived from either phenylalanine free amino acids (L-AAs) or a combination of low phenylalanine peptides with added amino acids (casein glycomacropeptide: CGMP). They are usually supplemented with vitamins, minerals and trace elements, and may contain essential and/or long chain fatty acids and prebiotics. In 1953, the first protein substitute was made using a low phenylalanine hydrolysed casein [2,3]; subsequently, the number and type of manufactured preparations have exponentially increased [4]. In 2008, CGMP, a by-product of whey from the manufacture of cheese, was introduced as an alternative protein substitute to L-AAs, but it is still unclear if this protein source has any advantage over conventional L-AAs in the dietary management of PKU. Overall, their composition, bioavailability and long term impact on metabolic efficacy has received limited systematic investigation in PKU.

This review examines the evidence of using the bioactive protein substitute CGMP compared to L-AAs in the treatment of PKU, focusing on benefits to blood phenylalanine stability, body composition, bone mass, density and geometry and the influence of protein substitutes on breath malodour and palatability.

## 2. Protein Substitutes Pharmacological Benefits

Protein substitutes meet the protein requirement for cellular function and growth and have several pharmacological and physiological functions (Table 1). They improve phenylalanine tolerance and optimise metabolic control by suppressing blood phenylalanine concentrations. This is particularly important during illness and trauma, where protein substitutes have a protective role by counteracting protein catabolism [5,6,7,8]. Irrespective of their nitrogen source, each protein substitute has a different amino acid profile consisting of essential and non-essential amino acids, and around 40% large neutral amino acids (LNAAs). They provide the principal source of tyrosine, although there is no consensus on the optimal amount required [9]. Similarly, there is no agreement on the quantity and ratio of branched chain amino acids, and there is also limited data about the absorption and retention of amino acids [10,11,12,13]. 

## 3. The Role of Functional Amino Acids in Protein Substitutes

Amino acids in protein substitutes have several nutritional, biochemical and physiological roles linked to growth, health and disease prevention [22,25]. Functional amino acids (essential or non-essential) regulate key metabolic pathways. They provide nitrogen, hydrocarbon skeletons and sulphur [26]; both nitrogen and sulphur are unable to be synthesised de novo. Some roles of functional amino acids include regulation of body composition and bone health, others include modulating bacterial flora, glucose homeostasis and inflammatory responses. Amino acids are also involved in cell signalling (including mammalian target of rapamycin complex 1 (mTORC1), and the interaction and generation of small peptides, glucagon-like peptide 1 (GLP-1), peptide-YY (PYY), serotonin and insulin. Insulin plays a key regulatory role in amino acid metabolism, and amino acids alter insulin action by regulating glucose and protein metabolism [27,28]. The composition of a protein substitute affects the rate at which amino acids are delivered into the systemic system, changing their cellular uptake and biological utilisation. Different rates of absorption have been reported when amino acids are ingested as free amino acids, peptides or bound to proteins [13,26,29]. Free amino acids appear in the peripheral plasma more quickly than those from an intact protein source [10]. Any protein substitute that can maximise amino acid absorption will increase anabolism and subsequently alter phenylalanine metabolism. 

## 4. What Is a Casein Glycomacropeptide (CGMP)?

In 1954 while working on a variant of *lactobacillus bifidus*, György et al. [30] found evidence of protein bound sialic acid (N acetylneuraminic acid) in cow’s milk. In 1965, Delfour et al. [31] established that this milk bound sialic acid protein was called κ casein and reported that CGMP was formed by separation of κ casein by the action of chymosin during cheese production. CGMP is found in the soluble whey elute [32] and constitutes 20–25% of total proteins in whey products manufactured from cheese whey. It is a 64 amino acid phosphoglycoprotein [33]. Five oligosaccharides (glycans) have been identified as part of the glycomacropeptide structure [32]. 

In its pure form, the glycophosphopeptide has an unusual amino acid sequence containing no aromatic amino acids (tryptophan, tyrosine, phenylalanine) or the sulphur amino acid cysteine [34]. Of the five glycan structures common to bovine CGMP, the one of most interest is the nine-carbon sugar molecule, sialic acid, which forms 7–9% of CGMP. This is a component of human milk oligosaccharides and neural tissues and is an integral part of brain gangliosides and glycoproteins. The glycan chains are attached via two types of glycosylation: *N*-linked when the glycan chain is attached to the amide side chain of the asparagine residue, and *O*-linked when the glycan is attached to the oxygen of a serine or threonine residue [35,36]. Around 60% of CGMP is glycosylated [37] with exclusively *O*-linking glycans. There is evidence to suggest glycosylation is a controlled hierarchical process that influences the associated biological activities of CGMP [38,39]. These bioactive properties provide a functional ingredient for the food and pharmaceutical industry.

## 5. Potential Clinical Properties of CGMP

Carbohydrates, whether free or bound to proteins or lipids, are essential communication molecules in inter and intracellular processes. The biological properties associated with CGMP include immunomodulatory, antimicrobial and prebiotic [32,35,40]. CGMP interacts with cholera toxins through the glycan chains [41,42], and bind to *E. coli* and *Salmonella enteritidis* [43]. It also has an important role in anticariogenesis; CGMP inhibits adherence of oral bacteria, preventing tooth decay [44,45]. In animal experiments, a CGMP enriched infant formula increased learning ability, which was linked to an increase in sialioprotein in the frontal brain cortex [46]. These findings need further investigation.

## 6. Potential Commercial Use of CGMP 

CGMP is an acidic peptide, highly soluble and heat stable [35]. It also has a wide pH range and solubility, and has emulsifying, gel and foaming properties, making it desirable in the food and nutritional products industry as it alters the structural matrix of foods and improves the texture and mouth feel. 

## 7. Adaptation of CGMP for Use as a Low Phenylalanine Protein Substitute in PKU

Isolating CGMP from cheese whey is difficult and expensive, with residual phenylalanine remaining in the final product [32]. CGMP has inadequate amounts of five indispensable amino acids: histidine, leucine, methionine, tryptophan, and tyrosine, but supplementation with these amino acids enables it to be used as an alternative to L-AAs [47]. 

The first case study using CGMP [47] was reported in a 29-year-old male with PKU. Over 15 weeks, CGMP and L-AA protein substitutes were compared. CGMP was supplemented with histidine, leucine and tryptophan providing 130% and tyrosine at 150% of the USA 2002 recommendation [48]. Added vitamins, minerals and trace elements were supplemented when taking CGMP. An additional 500 mg of tyrosine was taken orally twice daily, providing the same tyrosine intake as that from L-AAs. Significant increases in plasma glutamine, isoleucine, proline and threonine, with an overall increase in the LNAAs and a 16% increase in the BCAAs were noted. CGMP is naturally higher in threonine and isoleucine, explaining the observed increases. In a subsequent study in 2009, van Calcar et al. [49] compared the effects of L-AAs and CGMP in 11 subjects with PKU over 8 days. The CGMP product was supplemented with histidine, leucine, methionine and tryptophan, but the additional supplement of 1000 mg/day of tyrosine was omitted. This led to a mean fasting tyrosine concentration below the normal reference range in the CGMP group, with an expected increase in isoleucine and threonine consistent with the higher concentration in CGMP. After an overnight fast, plasma blood concentration of arginine, a conditionally essential amino acid, was significantly lower. The limiting amino acids added to the CGMP, histidine, leucine methionine and tryptophan, remained within the normal biochemical reference ranges, but tyrosine and arginine concentrations required further supplementation. Methionine supplementation was stopped as there was an adequate amount in the CGMP to meet the new lower requirements as suggested by Humayun et al. [50].

## 8. The Impact of CGMP on Blood Phenylalanine Control in PKU

Ten published studies have investigated the effect of CGMP compared to L-AAs on blood phenylalanine control. The majority (*n* = 7/10, 70%) have suggested no significant alteration in blood phenylalanine concentrations despite residual phenylalanine being present in CGMP [49,51,52,53,54]. Nine of ten studies reported higher blood phenylalanine concentrations when using CGMP, but only three studies demonstrated a statistically significant increase. All three studies were in children from one centre, but this included two long term longitudinal studies over 6 and 12 months [55,56], and one randomised controlled study over 6 weeks [57]. Four other studies collected data mainly in adults for a minimal period of 8 to 21 days, with suboptimal blood phenylalanine concentration at study baseline; some subjects were taking adjunctive sapropterin treatment that improved phenylalanine tolerance. Two studies were retrospective reviews in 11 teenagers and adults, with follow up at 20 and 29 months [58,59]. One study [54] examined CGMP as a food (GMP soft cheese) supplement in children; it was consumed 3 times daily over 9 weeks. No information was provided on its residual phenylalanine content or amino acid profile. The supplement was provided in combination with L-AAs and provided 50% of the total protein substitute intake. 

It is difficult to interpret the effectiveness of results from short-term studies. One of the earliest studies [49] suggested that the residual phenylalanine in the CGMP was too high at 0.4 g/100 g of product. This was only given to three subjects, all with high phenylalanine tolerance. In the remaining nine subjects, the CGMP composition was refined, with a phenylalanine content of 0.2 g/100 g of product. A statistically significant increase in blood phenylalanine was only evident in the longitudinal studies in children, with blood phenylalanine being maintained within a narrow therapeutic target range of 120 to 360 µmol/L. This suggests caution is necessary when using CGMP that contains residual phenylalanine, particularly in children with classical PKU. Table 2 lists the PKU studies using CGMP and their outcomes. The impact of residual phenylalanine may be less important in patients using adjunct drug management that improves phenylalanine tolerance or in teenagers and adults who maintain blood phenylalanine levels under a higher upper therapeutic target. Further studies are needed in adults and in pregnancy when CGMP is the only protein substitute source. 

## 9. Kinetic Properties of Protein Substitutes

There is evidence from animal studies that protein substitutes engineered to slowly release amino acids have improved physiological functions, but proving this remains a challenge in PKU [61]. The speed of absorption of dietary amino acids by the gut varies according to the type of ingested dietary protein. Whey protein is established as a ‘fast’ protein and casein as a ‘slow’ protein, the latter provides greater nitrogen retention and whole-body protein anabolism [62,63]. L-AAs are incapable of replicating the physiological actions of whole protein being directly absorbed from the small intestine [22]. Amino acids from L-AAs are rapidly absorbed, peak but then fall rapidly compared to amino acids slowly released from whole protein, and this influences their utilization [12,64,65]. Herrmann et al. [66] demonstrated that ingestion of large doses of L-AAs increased amino acid oxidation and nitrogen excretion, decreasing their availability for cellular functioning. For effective protein synthesis, all essential amino acids must be available to the tissues in appropriate amounts simultaneously [29]. There is circumstantial evidence to suggest that CGMP lowers the rate of amino acid absorption and improves nitrogen retention. Van Calcar et al. studied 11 subjects with PKU over 4 days and reported lower blood phenylalanine after an overnight fast using CGMP compared to L-AAs, implying a slower release of amino acids in CGMP. Two-hour post prandial blood urea nitrogen concentrations were lower, and insulin concentrations were marginally but significantly higher in the CGMP group, suggesting lower nitrogen excretion and improved amino acid utilisation. Any protein substitute that will imitate the physiological absorption of whole protein will theoretically improve growth, body composition and bone density, and may possibly influence inflammatory responses and appetite.

There are no kinetic studies reviewing the action of L-AAs versus CGMP on blood urea nitrogen, insulin or amino acid absorption. Until studies are reported, it cannot be concluded that CGMP improves amino acid utilisation. However, CGMP does influence phenylalanine and tyrosine variability over a 24-h period. In a randomised controlled crossover study [57], children with PKU were randomised to three groups taking CGMP or L-AAs as a protein substitute: group R1 (no dietary adjustment with CGMP), group R2 (dietary adjustment with phenylalanine from CGMP deducted from the dietary phenylalanine allowance) and group R3 (no dietary adjustment with L-AAs). Each arm of the study was for 14 days, and on the last 2 days, subjects had 4-hourly day and night blood spots measuring blood phenylalanine and tyrosine. All median phenylalanine concentrations were within recommended target ranges, there was a significant difference in median phenylalanine at each time point between R1 and R2 (*p* = 0.0027) and R1 and R3 (*p* < 0.0001), but no differences between R2 and R3. Tyrosine was significantly higher in the CGMP groups. This work shows two main findings: the residual phenylalanine given in R1 increased blood phenylalanine concentrations (in this group, 18% had phenylalanine concentrations greater than the target reference range compared to none in the R3 group), and secondly, CGMP appears to give less blood phenylalanine variability when compared to L-AAs. Any mechanism that permits a constant delivery of amino acids would allow a steady state of protein synthesis, improving body protein balance and skeletal muscle protein synthesis.

In a preliminary investigation [67] to review if CGMP compared to L-AAs altered pre and post prandial amino acid profiles in children with PKU, quantitative amino acids were measured after an overnight fast and 2 h post prandially after consuming breakfast and 20 g protein equivalent from the allocated protein substitute. CGMP was provided as CGMP1, in which the amino acid profile met WHO recommendations, or CGMP2, which had higher concentrations of histidine, tyrosine, tryptophan and valine. Forty-three children, median age 9 years (range 5–16 years) were studied; 11 took CGMP1, 18 CGMP2 and 14 L-AAs. The results showed, regardless of the protein substitute source, there was a significant increase in post prandial amino acids. In CGMP2, post prandial histidine (*p* < 0.001), leucine (*p* < 0.001) and tyrosine (*p* < 0.001) were higher than in CGMP1 (reflecting the additional amounts in this formulation), and leucine (*p* < 0.001), threonine (*p* < 0.001) and tyrosine (*p* = 0.003) were higher in CGMP2 than in L-AAs, reflecting the amino acid composition of the three different protein substitute formulations. There is a suggestion that CGMP does alter amino acid absorption, leading to a greater stability of phenylalanine over 24 h, but controlled kinetic studies are necessary. 

## 10. The Impact of CGMP on Growth and Body Composition in Children with PKU

In PKU, the impact of using a phenylalanine-restricted diet on physical growth was first reported in the late 1970s, and despite improvements in dietary treatment, contradictory findings on growth outcome are reported [68,69,70,71]. Early studies [72] demonstrated that children had improved growth if they were prescribed a protein equivalent from protein substitute that exceeded the WHO/FAO/UNU 1973 [73] safe levels of protein intake. Smith et al. [74] showed that even if amino acids are efficiently absorbed from the intestinal tract, there is a higher loss of nitrogen as urea when compared to natural protein. McBurnie et al. and Holm et al. [75,76] assessed height, weight and head circumference in two prospective collaborative studies, evaluating 133 and 124 children with PKU over 8 and 4 years, respectively. In both studies, weight and height increased similarly to that of control groups. 

In contrast, three European studies [77,78,79] found children with PKU had reduced height growth when compared to control subjects. Protein substitute intake was not always reported, but typical total protein intake only provided safe recommended intakes [73]. It is possible that phenylalanine deficiency may have occurred but was not described. Dhondt et al. [77] reported normal height and weight were achieved after dietary relaxation at 8 years of age. Schaefer et al. [78] reported negative weight and height in the first 2 years with catch up by 3 years of age. A recent systematic and meta-analysis examining growth in subjects with PKU [70] reported normal growth at birth and during infancy, but children were significantly shorter and had lower weight for age compared with reference populations during the first four years of life. Linear growth was reduced until the end of adolescence. These findings were not identified in patients with mild hyperphenylalaninemia on no dietary restrictions. 

Overall, optimal growth was noted in studies where total protein intake (a combined protein intake from natural protein and protein substitute) was higher [80,81,82,83]. Nitrogen balance is regulated by urea production [63,84], which is produced linearly in response to plasma amino acid concentrations. Ney et al. and Calcar et al. [49,85] suggested that CGMP may induce a slower and more sustained release of amino acids, leading to decreased urea and greater availability of amino acids for protein synthesis, possibly leading to improved growth. 

In PKU, it is important to monitor lean and fat mass, but there are no long-term prospective studies or systematic/meta-analyses describing body composition in PKU. Of eleven studies reported in children (Table 3), any comparison is challenging due to an absence of national reference standards, different body composition techniques, variable pubertal status and different PKU phenotypes. Of six controlled studies, compared with healthy controls, four showed no statistically significant differences in body composition. One study demonstrated a correlation with increased blood phenylalanine concentrations and higher fat mass in male subjects with PKU only [86]. Albersen et al. [87] showed body fat was significantly higher in subjects with PKU, and higher in females >11 years. Long-term associated comorbidities such as type II diabetes and cardiometabolic diseases may be linked to altered body composition, with evidence suggesting an association between abdominal obesity, increased insulin resistance and cardiovascular disease. Therefore, the composition of a protein substitute needs careful formulation as this may alter body composition and possibly long-term health outcomes [88,89,90].

To date, only two studies have examined the role of CGMP compared to L-AAs on body composition and growth in PKU: one three-year prospective study [99] in children, and a retrospective review in adults by Pena et al. in 2021. In the three-year study, *n* = 19 children (median age 11 years; range 5–15 years) took L-AAs only, *n* = 16 (median age 7.3 years; range 5–15 years) took a combination of CGMP and L-AAs (CGMP50), and *n* = 13 (median age 9.2 years; range 5–16 years) took CGMP only (CGMP100). A dual-energy X-ray absorptiometry (DXA) scan at enrolment and 36 months measured lean body mass (LBM), % body fat (%BF) and fat mass (FM). Height was measured at enrolment, 12, 24 and 36 months. No correlation or statistically significant differences (after adjusting for age, gender, puberty and phenylalanine blood concentrations) were found between the three groups. The change in height z-scores (L-AAs 0, CGMP50 +0.4 and CGMP100 +0.7) showed a trend that children in the CGMP100 group were taller, had improved LBM with decreased FM and %BF, although this did not reach statistical significance. We can only speculate about this suggested trend shown in the CGMP100 group. One possibility is that the branched-chain amino acids leucine and isoleucine (the latter is naturally higher in CGMP) modulate protein turnover, as both are potent modulators of insulin and glucose metabolism [100]. If insulin sensitivity is enhanced, it is possible that growth could be improved. Further long term studies are needed to confirm these findings. 

## 11. Impact of CGMP Compared to L-AAs on Bone Mass, Density and Geometry in Children with PKU

Bone mass is maintained by a complex and dynamic process involving resorption of bone by the osteoclast and formation of bone by the osteoblast. In children, this is a dynamic continuous process of modelling and remodelling [101]. Peak bone mass, which programmes the future risk of osteoporosis, is established in childhood and adolescence [102,103]. Factors that influence bone mass include genetics, lean mass, adiposity, adipocytokines, physical activity and nutrition. The relationship between fat and bone is contentious. Evidence [103] suggests that in early childhood, obesity confers a structural advantage, but with age this relationship is reversed, and excessive fat is detrimental. Clark et al. [104] in 3082 healthy children, reported a positive relationship between adiposity and bone mass accrual. Others have reported conflicting findings [105,106]. Lean body mass is the strongest significant predictor of bone mineral content [107,108] and relates to bone mass and skeletal development in children.

Dietary protein promotes peripubertal bone growth and slows bone loss [109]. Protein is necessary for optimal bone metabolism during growth, positively influencing bone mass, density and strength [109,110,111]. In children and adults with PKU, bone density is inconsistently reported [112,113,114,115,116,117,118]. Four systematic and three meta-analysis studies report mixed results. Enns et al. reported nine suboptimal bone health outcomes. The scope of this review was on general health problems in PKU, and therefore it failed to interpret the results on bone health in depth. Hansen et al. described a lower spine bone mineral density, but this review had methodological errors and assessment bias. Demirdas et al. [119] reported bone mineral density (BMD) was within the normal range; although it was lower than normal, it was not clinically significant. There was no correlation with phenylalanine concentrations, vitamin D, parathyroid hormone and individual nutrients. De Castro et al. [120] supported the findings from Demirdas et al., showing BMD was lower than that of the reference groups but within the normal range. They also demonstrated an imbalance between bone formation and resorption, favouring bone removal. 

Solverson et al. [121] studied the effect of three different diets on bone strength in mice with or without PKU. They were given a low-protein diet with (a) CGMP, (b) L-AAs or (c) a normal (casein) diet. The PKU mice fed either CGMP or L-AAs had a lower BMD compared with non-PKU mice. In PKU mice fed the L-AAs, the femur length independent of gender was significantly shorter compared to that of the PKU mice given CGMP or a normal diet. Skeletal fragility (brittle and weak femora) was a consistent finding in the PKU mice regardless of gender or diet. The reduction of BMD and bone mineral content (BMC) of the femora measured by DXA was more pronounced in the mice receiving L-AAs compared to those receiving CGMP. This group concluded that the type of protein influenced bone outcome in mice, with CGMP giving better results compared to L-AAs. However, careful consideration is needed to determine the impact of CGMP or L-AAs on bone growth. In humans, bone growth is a slow, multifaceted process affected by hormonal patterns, gender, obesity, dietary intake and physical activity.

Only one three-year longitudinal study [122] in children with PKU has compared the impact of CGMP and L-AAs on bone mass, density and geometry (comparing the same group of children who participated in the body composition study previously described). Measurements were taken by DXA and peripheral quantitative computer tomography (pQCT), in addition to blood biochemistry and bone turnover markers. No statistical significance was evident between the three study groups (L-AAs, CGMP50 or CGMP100). In all three groups, there was a strong positive correlation between bone resorption and formation markers: type 1 collagen cross-linked C telopeptide (β CTX) and procollagen type 1 terminal propeptide (P1NP), and there was evidence of an increased PINP in the CGMP100 group independent of age compared to the L-AA group (*p* = 0.04). The synergy between bone formation and resorption shows active bone turnover and reflects appropriate bone growth since these markers are derived from physiological processes. Bone density was clinically normal, although the median z-scores were below the population mean and agreed with the findings of systematic reviews by Demirdis et al. and de Castro et al. Bone remodelling processes appeared active in children with PKU taking either L-AAs or CGMP, but it was unknown why the median z-scores were below the population norm.

## 12. Does Glycomacropeptide Improve Palatability of Protein Substitutes?

A potential advantage of using a peptide-based protein substitute is the altered taste profile. L-AAs are generally bitter tasting, and both children and adults dislike the aftertaste they leave post consumption [123]. In a blind sensory study, Lim et al. 2007 evaluated the acceptability of CGMP compared to L-AAs and found CGMP was rated favourable for odour and taste. This improved taste profile has been observed by other researchers [49,51,52,54,55,59,124]. Pena et al. [53] highlighted the lack of uniformity in the methods used to evaluate palatability, with some studies evaluating food and others liquid based CGMP protein substitutes. The improved taste profile may improve concordance with a lifelong rigorous diet.

## 13. Impact of CGMP on Breath Malodour in Children with PKU

In clinical practice, caregivers of children with PKU report their children have breath malodour, particularly after protein substitute consumption. This may increase non adherence by lowering self-esteem and affect interpersonal communication, leading to social isolation. No study has quantitatively measured breath odour in children with PKU. In a randomised, crossover study using gas chromatography ion mobility spectrometry (GS-IMS), exhaled volatile organic compounds were measured in children taking CGMP or L-AAs over the course of 10 h [123]

Forty children (20 PKU; 20 healthy non-PKU controls) were recruited; the children with PKU took either L-AAs or CGMP exclusively for one week in a randomised order. On the seventh day, seven exhaled breath samples were collected over a 10-h period. Subjects than transferred to the alternative protein substitute for a week, and the breath sampling process was repeated. In the PKU group, the aim was to collect breath samples 30 min after consuming their protein substitute; this happened in all but three cases, when breath samples were collected 5 min after protein substitute consumption. In all three groups (L-AAs, CGMP and controls), fasting breath samples contained similar numbers of volatile organic compounds (VOCs) (10–12). Similarly, post prandial samples showed no significant differences in the number of exhaled VOCs (12–18) between L-AAs/CGMP and controls, or between L-AAs and CGMP. A different breath signature occurred in the three subjects who had breath measurements 5 min post completing their protein substitute. In this subset, a higher number of VOCs (25–30) were detected; however, these were no longer detectable at 30 min post consumption. This study demonstrated that protein substitutes have a transient effect on exhaled breath, and after 30 min post consumption, VOCs in children with PKU were no different to those of controls. Timing food and drink with protein substitute consumption may potentially reduce or eliminate the immediate unpleasant protein substitute breath odour.

## 14. Summary

In PKU, evidence suggests that the use of a bioactive CGMP protein substitute does not show any overwhelming benefit compared to L-AAs on post prandial amino acid absorption, body composition, bone mineral density or breath odour. It is clear that CGMP increases blood phenylalanine concentrations, particularly in children with a low phenylalanine tolerance. However, there is a trend that children taking CGMP as their sole source of protein substitute are taller, with improved lean body mass and decreased fat mass. Overall, the residual phenylalanine content in CGMP appears to be a limitation, particularly for those with minimal or no phenylalanine hydroxylase activity. The full clinical potential of CGMP in PKU has not yet been determined, and its role in gut microbiota and potential brain development awaits further investigation.

## Figures and Tables

**Table 1 nutrients-14-00807-t001:** Functional properties of protein substitutes in PKU.

Functional Properties	Action	References
Large neutral amino acids (LNAAs)	Phenylalanine transport from the plasma into the brain is via the LNAA transporter (LAT1). Competition at the blood brain barrier using LNAAs for LAT1 prevents excess phenylalanine from entering the brain, preventing neurocognitive damage	[14,15,16]
	LNAAs and cationic amino acids cross the intestinal mucosa via a carrier protein system. The affinity of the amino acids for the intestinal carrier is higher than at the blood brain barrier. By providing LNAAs, there is a decreased entry of phenylalanine across the intestinal mucosa	[17,18,19]
Normal growth and cellular function	Provide nitrogen to maintain and improve muscle mass and promote growth	[20,21]
Provide a source of nitrogen for the synthesis of nitrogen containing compounds	Nitrogen is necessary for the manufacture of small molecular substances, e.g., nitric oxide	[22]
Provide tyrosine	Phenylalanine to tyrosine conversion is severely limited or absent in classical PKU. Tyrosine becomes a surrogate essential amino acid, and adequate amounts must be provided by protein substitutes to prevent deficiency. Tyrosine is important for the biosynthesis of neurotransmitters, thyroxine and melanin	[8,9]
Optimise blood phenylalanine control	Protein substitutes support stabilisation of blood phenylalanine concentrations by providing a complement of amino acids (except phenylalanine) allowing protein anabolism and nitrogen retention. For maximum effectiveness, they must be given frequently throughout the day	[7,23]
Prevent nutritional deficiencies	Most protein substitutes are supplemented with vitamins, minerals and trace elements. Adherence with separate vitamin and mineral supplements is poor in patients with PKU	[24]

**Table 2 nutrients-14-00807-t002:** Studies using CGMP compared to L-amino acid protein substitutes in PKU.

Author/ Year	Country	Study Design Age (Range)	Nos of Subjects/ Gender	PKU Phenotype	Study Intervention	Mean/Median Phenylalanine Concentrations in L-AAs Compared to CGMP (μmol/L)
Van Calcar [49] 2009	United States	Cross-sectiona l23 y ± 7 (11–31)	11 4 F, 7 M	10 Classical 1 Variant	100% L-AAs vs. 100% CGMP 4 days on each product	L-AAs = 619 CGMP = 676, *p =* ns
MacLeod [60] 2010	United States	Cross-sectional 23 y ± 7 (11–31)	11 4 F, 7 M	11 Classical	100% L-AAs vs. 100% CGMP 4 days on each product	L-AAs = 619 CGMP = 676, *p =* ns
Ney [52] 2016	United States	Randomised crossover clinical study (15–49)	301 8 F,12 M	20 Classical 10 Variant	21 days: 100% CGMP or 100% L-AAs	L-AAs = 655 CGMP = 777, *p =* ns
Zaki [54] 2016	Egypt	Clinical study 6.7 y (5.0–11.8)	10 4 F, 6 M	10 Classical	9 weeks: 50% CGMP + 50% L-AA 9 weeks: 100% L-AA	100% L-AA s = 490 CGMP 50% + 50% L-AAs = 376, *p =* ns
Pinto [59] 2017	Portugal	Retrospective longitudinal study 27 y ± 10 (13–42)	11 8 F, 3 M	6 Classical 4 Mild 1 HPA	Median 20 months: *n* = 11 CGMP, *n* = 11 L-AAs	L-AAs = 516 CGMP = 540, *p =* ns
Daly [55] 2017	UK	Prospective clinical study 11 y (6–16)	21 9 F, 12 M	20 Classical 1 Mild	6 months *n* = 12 CGMP *n* = 9 L-AAs	L-AAs: pre study 325, end of study 280, *p* = ns CGMP: pre study 275, end of study 317, *p* < 0.02
Ahring [51] 2018	Denmark	Randomised crossover clinical study. 4 PS given over 4 visits 33.3 y ± 11.2 (15–48)	8 7 F, 1 M	8 Classical	PS1 = CGMP, PS2 = L-AAs PS1 and PS2 same AA profilePS3 = CGMP + L-AAs, PS4 = L-AAs PS3 and PS4 same L-AA profile but no Phe	L-AAs = 688 CGMP = 819, *p* = ns
Daly [56] 2019	UK	Prospective clinical study over 12 months 9.2 y (5–16)	48 21 F, 27 M	46 Classical 2 Mild	12 months *n* = 29 CGMP *n* = 19 L-AAs	L-AAs pre study 315, 52 weeks 340, *p* = 0.236 CGMP pre study 270, 52 weeks 300, *p* = 0.001
Daly [57] 2019	UK	Randomised control study (RCT) 10 y (6–16)	18 11 F, 7 M	17 Classical 2 Mild	6-week RCT 2 weeks CGMP 100% no dietary changes (R1) 2 weeks CGMP 100% minus dietary phenylalanine contributed from CGMP (R2) 2 weeks L-AAs nodietary changes (R3)	Median phenylalanine R1: 290 (30–580) R2: 220 (10–670) R3: 165 (10–640) R1 vs. R2, R1 vs. R3 *p* < 0.0001 R2 vs. R3, *p =* 0.0009
Pena [58] 2021	Portugal	Retrospective longitudinal study 28 y (15–43)	11 8 F, 3 M	3 Classical 3 Late diagnosed 3 Mild 2 HPA	29 months CGMP 66%, L-AAs 34% *n* = 4 CGMP 100% *n* = 4 CGMP 50 < 100% *n* = 2 CGMP < 50%	Pre study on L-AAs: 562 ± 289 Post study L-AAs and CGMP 628 ± 317, *p =* ns

Legend: PKU phenylketonuria; L-AA, amino acid protein substitute; CGMP, caseinglycomacropeptide; PS, protein substitute; ns, not significant; HPA, hyperphenylalaninemia; F, female; M, male; y, years; m, months; vs, versus.

**Table 3 nutrients-14-00807-t003:** Studies measuring body composition in children with PKU.

Author/Year	Number/Age of Subjects Body Composition Measurement Technique	Parameters Measured	Main Outcome	Limitations
Allen 1996 [91] Australia	*n* = 30 PKU (classical) Mean age: 9.6 y *n* = 65 control Mean age: 11.2 y Skinfold thickness	Body fat Resting energy expenditure	No differences in body fat compared to controls No difference in resting energy expenditure	Skinfold measurements provide no information on lean mass.
Dobbelaere 2003 [68] France	*n* = 20 PKU (classical) *n* = 20 control Mean age: 4.5 y Age- and gender-matched Skinfold thickness Bioelectrical impedance	Weight, height, body mass index (BMI) head circumference Skin folds triceps, biceps, subscapular and suprailiac measurement Body density, body fat, lean mass Blood tyrosine and phenylalanine concentrations Zinc, selenium, thyroid, insulin, growth factor Weighed 4-day dietary intake	No differences in body composition compared with controls Growth was significantly different from that of reference population *p* < 0.05 No correlation with phenylalanine biochemical bloods or calorie intake	Body mass index measures nutritional status, not body composition Impedance associated with poor accuracy for individuals and groups
Huemer 2007 [92] Study over 12 months Austria	*n* = 34 PKU (classical) *n* = 34 control Mean age: 8.7 y Age/gender-matched Total body electrical conductivity (TOBEC)	Weight, height, BMI % fat, fat-free mass Blood phenylalanine concentrations	No differences between groups for the measured parameters Significant correlation between natural protein g/kg/d and fat-free mass	TOBEC rarely used and unknown accuracy compared to other body composition measurements
Albersen 2010 [87] The Netherlands	*n* = 20 PKU (classical) *n* = 20 control Mean age: 10 y Age/gender-matched BodPod/whole-body air displacement plethysmograph	Weight, height, BMI % body fat Blood phenylalanine concentrations	No difference for weight, height, BMI Body fat significantly higher in PKU despite similar BMI to that of controls *p* = 0.002 Body fat higher in girls >11 y, *p* = 0.027 Body fat increased with weight only in PKU No correlation with blood phenylalanine	4/20 PKU children were from different ethnic background
Adamczyk 2011 [93] Poland	*n* = 45 PKU (classical) Mean age: 13.8 y Group 1 = 15 prepubertal Group 2 = 18 pubertal good control Group 3 = 12 pubertal poor control Dual X-ray absorptiometry (DXA)	Weight, height, BMI Lean body mass Fat mass Total bone density Bone mineral content Ratio of bone mineral content/lean body mass Bone markers Data compared with Polish DXA reference values	Normal body fat and lean body mass Statistically significant differences for ratio of bone mineral content/lean body mass between groups Blood phenylalanine negatively affected bone status	No control group DXA radiation exposure, whole-body bias dependent on size, gender and amount of fat
Douglas 2013 [94] USA	*n* = 59 PKU (classical and mild) Mean age: 14.4 y BodPod/whole-body air displacement plethysmograph Tricep, subscapular, suprailiac, thigh skinfold	Weight, height, BMI Body fat	Normal body fat Lean mass not evaluated Inverse relationship between age and body fat *p* = 0.016	Mixed PKU phenotype No control group Agreement between skinfold depends on equations used to convert measurement to body fat
Rocha 2012 Rocha 2013 [95,96] Portugal	*n* = 89 PKU (classical, mild, hyperphenylalaninemia) Mean age: 14.4 y *n* = 78 controls Mean age: 15.9 y Bioelectrical impedance analysis (BIA)	Weight, height, BMI Fat mass Lean body mass Body cell mass Muscular mass Phase angle	No differences in fat mass No differences in lean body mass No differences in body cell, muscular mass or phase angle All classical PKU negative height z-score No differences in height compared to controls in children aged <19 y In PKU group, aged >19 y, height statistically significantly worse than that of controls *p* = 0.017	Impedance is associated with poor accuracy for individuals and groups Mixed PKU phenotype
		Blood pressure, amino acids Glucose, insulin Total cholesterol, high-density cholesterol Triglycerides, C- reactive protein, uric acid Assessment of protein substitute and natural protein intake	Anthropometric parameters no differences to controls Higher triglycerides/high density cholesterol in PKU group Metabolic syndrome no difference compared with controls In PKU subjects, those with central obesity had significantly higher triglycerides/high-density cholesterol compared to those without central obesity	
Doulgeraki 2014 [97] Greece	*n* = 48 PKU (classical) Mean age: 10.9 y 32 HPA (mild hyperphenylalaninemia) Mean age: 10.9 y *n* = 57 control Age/gender-matched Dual X-ray absorptiometry (DXA)	Lean body mass Fat mass Bone mineral density	No differences in body composition Weight and BMI significantly different between mild PKU and classical PKU Bone mineral density lower in classical PKU compared to mild and controls Fat mass significantly higher in PKU teenagers with poor phenylalanine control Positive correlation between bone, muscle and fat mass in both groups and fat mass and phenylalanine concentrations	Mixed PKU phenotype Control group not reported in study DXA radiation exposure, whole-body bias dependent on size, gender and amount of fat
Mazzola 2016 [98] Brazil	*n* = 27 PKU *n* = 11 early diagnosed *n* = 16 late-diagnosed (classical and mild) *n* = 27 control Mean age: 12 y Age/gender-matched Bioelectrical impedance analysis (BIA)	Weight, height, BMI Fat mass Lean body mass Extracellular mass/body cell mass ratio Phase angle (PA)	No differences in body fat No differences in lean body mass No effect on time of diagnosis or PKU phenotype	Age at diagnosis variable, some early and late-treated PKU Mixed PKU phenotype
Sailer 2020 [86] USA	*n* = 30 PKU *n* = 30 control Mean age: 11.6 y Age/gender-matched 4 subjects on Kuvan Dual X-ray absorptiometry (DXA)	Weight, height, BMI Fat mass Lean body mass 24 h dietary recall	Male subjects with PKU had significantly lower lean body mass and more fat mass compared to controls *p* = 0.024 No differences for females and controls when measuring same parameters Age/fat mass positively correlated with blood phenylalanine, *p* = 0.02 Protein substitute negatively correlated with blood phenylalanine *p* = 0.04 Males with PKU had significantly lower height compared with controls *p* < 0.05 No difference in energy intake between the groups	Mixed PKU phenotype 13% on sapropterin DXA radiation exposure, whole-body bias dependent on size, gender and amount of fat
Daly 2021 [99] UK	*n* = 48 PKU Mean age: 9.2 y (5–16) 3 groups taking different protein substitutes *n* = 19 L-AAs only *n* = 16 CGMP and L-AAs (CGMP50) *n* = 13 CGMP only (CGMP100) Dual X-ray absorptiometry (DXA)	Weight, height, BMI Fat mass Lean body mass % body fat	No correlation or statistically significant differences (after adjusting for age, gender, puberty and blood phenylalanine concentrations) were found between the groups for fat mass, % body fat or lean body mass The change in height z-scores: L-AAs 0, CGMP50 +0.4, CGMP100 +0.7 showed a trend that children in the CGMP100 group were taller, had improved lean body mass with decreased fat mass and % body fat	DXA radiation exposure, whole body bias dependent on size, gender and amount of fat No control non PKU group Different intake of CGMP protein substitute

Legend: Sapropterin, drug treatment for PKU; BMI, body mass index; PKU, Phenylketonuria; L-AA, amino acid protein substitute; CGMP, caseinglycomacropeptide protein substitute; PS, protein substitute; ns, not significant; F, female; M, male; HPA, hyperphenylalaninemia; y, years

## Data Availability

Data available at the request of the corresponding author.

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
