# Peer review of "Glycomacropeptide in PKU—Does It Live Up to Its Potential?"

_nutrients, 2022, doi:10.3390/nu14040807_

Round 1

Reviewer 1 Report

This is a well-done manuscript that brings new information regarding a diet therapy and nutritional manegement of Phenylketonuria.

Abstract is presented in a clear, succinct and direct way. Introduction is clear, containing all the necessary information to understand the objectives of this study. Methodology is clear and straightforward. Results and discussion are clear and in accordance with the objective.

Author Response

Dear Reviewer and Editor 

Thank you for your review, please see the response.

Reviewer 2 Report

This is a very interesting review on the use of Glyconacropeptide (CGMP) vs Mono Amino Acids (L-AA) in the treatment of phenylketonuria.

The paper is well structured, the reasoning is sounding and the reference are appropriate.

We don’t have any major concerns for the paper.

Author Response

Dear Reviewers and Editor

Thank you for your review. We have made suggested changes.

Please see the response 
